# Biomechanical optimization of iliolumbar fixation strategies for unilateral vertical sacral fractures: Prioritizing stability-mobility balance via finite element analysis

Yupeng Ma[1,2], Weiwei Liu[1], Tao Huang[1], Huanyu Hong[1], Yong Zhao[2,3], Guofeng Xu[4]*, Yu Li[1]*

**1** The First Ward of Trauma Orthopedics, Yantai Shan Hospital, Yantai, Shandong Province, People's Republic of China, **2** Yantai Key Laboratory for Repair and Reconstruction of Bone & Joint, Yantai Shan Hospital, Yantai, Shandong Province, People's Republic of China, **3** Orthopaedics Department, Yantai Shan Hospital, Yantai, Shandong Province, People's Republic of China, **4** Yantai Zhifu District Center for Disease Control and Prevention, Shandong Yantai, Shandong Province, People's Republic of China

* dclyxinger@hotmail.com (YL); 314088136@qq.com (GX)

## Abstract

### Objective

This study aims to optimize iliolumbar fixation strategies for unilateral vertical sacral fractures via finite element analysis, by comparing stability, implant stress, and lumbar mobility to identify the optimal clinical option.

### Methods

A finite element model of the pelvis and L3-L5 lumbar spine was constructed to simulate four fixation models (L4L5IS, L5S1IS, L5IS, S1IS) under a 600 N vertical load. Sacral vertical displacement, implant stress, and fracture line separation were analyzed.

### Results

Double-segment fixation (L4 + L5 + iliac screw and L5 + S1 + iliac screw) demonstrated superior sacral stability compared to single-segment fixation (L5 + iliac screw and S1 + iliac screw). The L5 + S1 + iliac screw configuration achieved the best balance of stability and lumbar mobility. Stress concentrations were primarily observed at iliac screw connectors, but all models remained within safe mechanical limits.

### Conclusions

Double-segment fixation, particularly the L5 + S1 + iliac screw model, is recommended for optimal sacral stability. For cases with compromised S1 pedicles, L4 + L5 + iliac screw fixation is a reliable alternative. Short-segment fixation is viable when prioritizing lumbar mobility.

**Data availability statement:** All relevant data are within the manuscript and its Supporting information files.

**Funding:** The author(s) received funding support from the National Natural Science Foundation of China (No. 81641171 & No. 81301553), Shandong Provincial Key R&D Program of China (No. 2018GSF118064), Medical and Health Technology Development Program of Shandong Province, China (No. 202104070173 & No. 202404070931), Distinguished Middle-Aged and Young Scientist Encourage and Reward Foundation of Shandong Province, China (No. BS2013SF015), Science & Technology Innovation Development Project of Yantai City, China (No. 2021MSGY049 & No. 2021YD045 & No. 2022YD048), Binzhou Medical University "Clinical+X" Scientific and Technological Innovation Project (No. BY2021LCX32), and Binzhou Medical University 2025 Clinical Medicine Plus X Project (No. 2025CMX1001).

**Competing interests:** The authors have declared that no competing interests exist.

## 1. Introduction

Sacral fracture is a common and complex type of posterior pelvic ring injury, and its treatment has been an important research direction in the field of trauma orthopedics. Especially in osteoporotic patients, it seriously affects the quality of life and recovery ability of patients [1–3]. It has been shown that there are various treatment methods for sacral fractures, among which iliolumbar fixation, sacroiliac screws and triangular fixation modalities are commonly used in the clinic [4–7].

Kach and Trentz [8] were the first to apply the pedicle screw system of the spine to the ilium, a procedure they termed "iliolumbar fixation" (also referred to as lumbopelvic/spinopelvic fixation). It has been observed that iliolumbar fixation exhibits robust postoperative stability, attributable to the lengthy screws positioned between the iliac bone cortices, which anchor the implant within the pelvic ring with greater stability than the osteoporotic cancellous bone of the sacrum. This configuration effectively transfers forces from the ilium to the lumbar spine [9].

Iliolumbar fixation is a well-established technique with a high degree of flexibility in clinical application. Despite the fact that the sacroiliac screw technique is more minimally invasive and has been suggested to offer superior stability when used in combination with triangular fixation patterns, iliolumbar fixation remains a valuable option, particularly in cases of difficult reduction. This is especially true for individuals with restricted access to the sacroiliac region, for whom iliolumbar fixation provides a reliable alternative [5,10].

The specific biological issue of interest in this study is the effectiveness of internal fixation and its stability and stress distribution under different modes of fixation.There have been many biomechanical studies on sacral fracture fixation, most of these studies have focused on sacroiliac screw and triangular fixation techniques, and studies related to iliolumbar fixation have tended to be comparisons between these two techniques [4,5,11–14]. Considering that in clinical work if the sacroiliac screw channel is damaged or limited by medical conditions and levels iliolumbar fixation is still a good option in sacral fracture treatment. There is a lack of relevant biomechanical studies. This study aims to address this knowledge gap by investigating the biomechanical properties of different internal fixation models for iliolumbar fixation using finite element analysis, with the ultimate goal of developing more effective fixation strategies for clinical practice.

This study employs the finite element analysis (FEA) method, which integrates three-dimensional model construction with mechanical property simulation. The FEA method offers the advantage of accurately simulating the mechanical environment of sacral fracture and the effects of different fixation methods. By constructing a finite element model of sacral fracture, the stability and stress distribution of different internal fixation models can be investigated, thereby providing a scientific basis for clinical treatment.

This study primarily investigates the stability, stress distribution, and biomechanical characteristics of the sacrum under various iliac-sacral fixation models in unilateral longitudinal sacral fractures. It aims to establish a foundation for selecting optimal internal fixation strategies in clinical practice. Through this research, we seek

to propose novel theories to enhance the therapeutic efficacy of unilateral iliac-sacral fixation surgical approaches for patients with sacral fractures.

## 2. Methods

The study was approved by the Ethics Committee of Yantai shan Hospital and conducted in accordance with the ethical standards outlined in the Declaration of Helsinki. Volunteer recruitment for this study commenced on May 1, 2023, and concluded on December 31, 2023. Prior to study enrollment, all participants were provided with a detailed explanation of the study objectives, experimental procedures, potential risks, benefits, confidentiality measures, and their right to withdraw from the study at any time without repercussions. This ensured that participants had a comprehensive understanding of the study, and informed consent was obtained.Specifically, written informed consent was obtained from each participant, who signed a standardized consent form to document their voluntary agreement to participate in the study.

### 2.1. Construction of a finite element model

In this study, the pelvis and the L3-L5 segment of the lumbar spine of a healthy adult female (height 165 cm, weight 65 kg) were examined, and the image data were obtained using a 64-slice helical CT scanner (Philips; voltage 120 kV; current 100 mA). The CT image data were imported into Mimics 17.0 (Materialise, Belgium) in DICOM format for processing, with the objective of generating a 3D geometric model of the lumbar spine and pelvis (Fig 1). The slice layer thickness was 1 mm.

To ensure that the model accurately reflects the anatomical features, a CT gray value segmentation technique was used to distinguish the boundaries of different parts of the skeleton. Subsequently, the model was imported into 3-matic software (Materialise, Belgium), where the smoothing function was used to correct surface imperfections and optimize the structure of the model for subsequent manipulation and analysis.

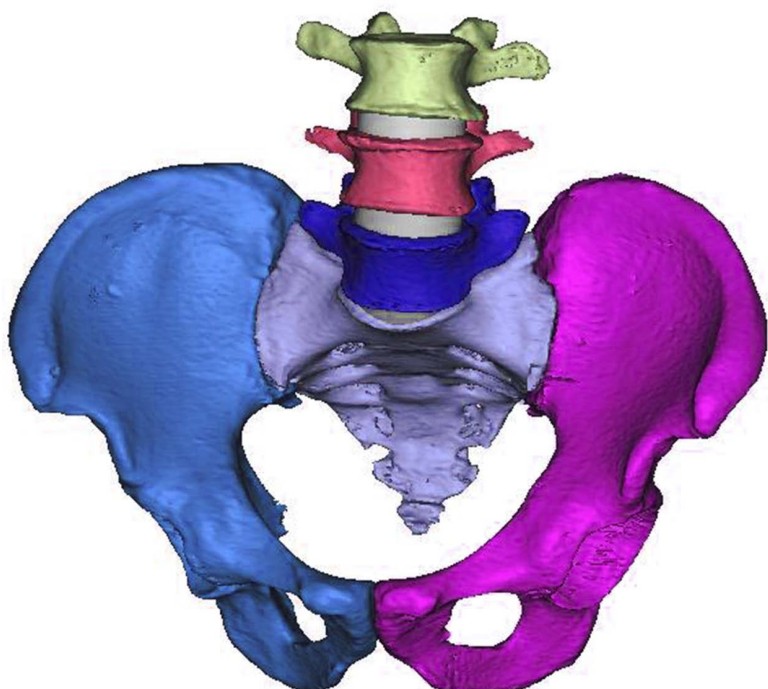

**Fig 1. A three-dimensional geometric model of the pelvic spine.**

In order to simulate an AO type C1.3 (Denis II) sacral fracture, the original sacral model was bisected along the sacral foramen, resulting in two parts that represent a typical posterior ring injury (Fig 2).

It is assumed in the model that the anterior ring damage is restored to stability after fixation, so as to exclude the interference of different anterior ring fixation methods on the results of the rear ring study. In order to further improve the FEA model, the Remesh module is used to delineate the mesh in 3-matic. The pelvic finite element model utilized 4-node linear tetrahedral elements (C3D4) for bony structures, with automatic mesh generation configured to a maximum element size of 5 mm to ensure uniform density control.This meshing density falls within the commonly adopted range for pelvic FE models reported in prior studies and validations [11,15–17]. As the primary aim was comparative biomechanics across fixation constructs, we prioritized a consistent mesh specification across models to ensure fair between-group comparison. The model consisted of a total of 239,733 elements and 72,520 nodes. Each node had 3 translational degrees of freedom (along the x, y, and z axes), resulting in a total of 217,560 degrees of freedom. The mesh model was then imported into Mimics for material property assignment, where the material was defined as non-homogeneous, isotropic, and linear elastic. The material parameters for specific bone components were determined using Mimics' built-in equations and CT gray value classification techniques. The CT value range for cancellous bone was set between 101 and 816 Hounsfield Units (HU) (the standard unit for CT image density measurement), while cortical bone was defined for CT values above 816 HU. These CT value ranges were determined based on CT grayscale value classification: in medical CT images, grayscale

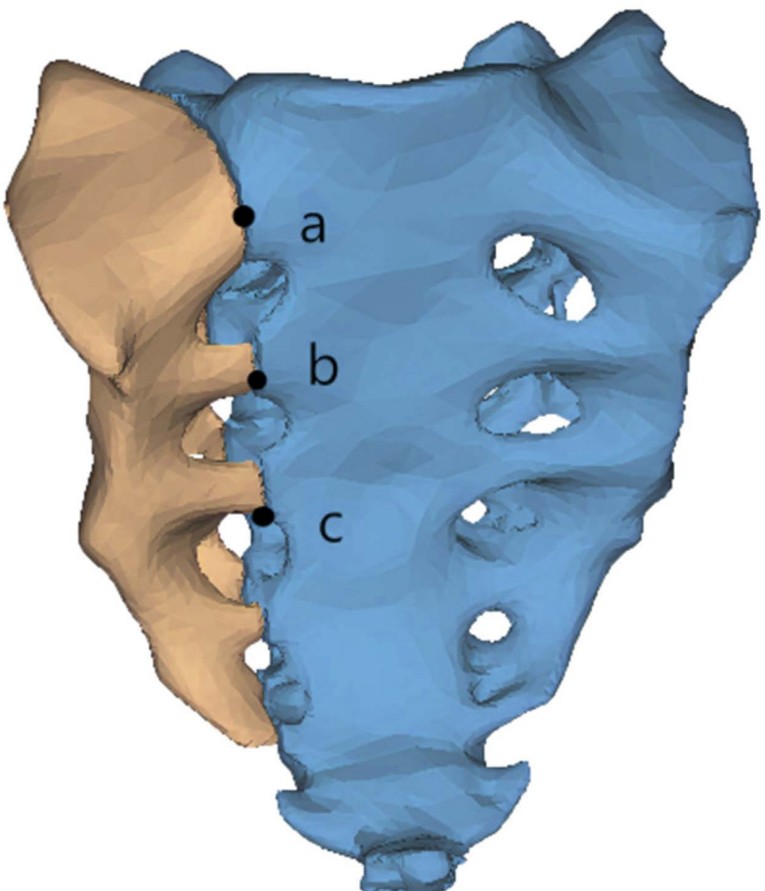

**Fig 2. Simulated longitudinal fracture of the sacrum.**

intensity directly correlates with CT values (HU), which reflect the X-ray attenuation coefficient of bone tissue (and thus its density). The division of cancellous and cortical bone CT value ranges (101–816 HU for cancellous bone, >816 HU for cortical bone) references established literature on pelvic finite element modeling [15,16], ensuring consistency with clinical anatomical and radiological standards. This approach eliminated the need to manually distinguish the boundary between cortical and cancellous bone, thereby reducing modeling time.This approach eliminated the need to distinguish the boundary between cortical and cancellous bone, thereby reducing modeling time. The relationship between bone density, elastic modulus, and CT values was then established using the following empirical equations [15,16]:

$$\rho = 0.047 + 1.122 \times 10^{-3}\,\mathsf{HU} \quad \mathsf{HU} \leq 816$$

$$\rho = 7.69 \times 10^{-4}\,\mathsf{HU} + 1.028 \quad \mathsf{HU} \geq 816$$

$$E = 1904\rho^{1.64} \quad (\text{Trabecular Bone})$$
$$E = 2065\rho^{3.09} \quad (\text{Cortical Bone})$$

In the equation, ρ and E represent the density and elastic modulus of a specific region of the bone tissue in the CT image, respectively, with units of g/cm³ and MPa. CT denotes the CT value at that specific point.The Poisson's ratio was set to 0.3 for cortical bone and 0.2 for trabecular bone [11].

The geometric models of the implants were created by SolidWorks 2017 (Dassault Systemes, USA) software. The implants included pedicle screws and iliac screws made of titanium alloy. Once modeling was completed, it was imported into 3-matic software, assembled with the pelvic model and a mesh was generated, followed by assigning material properties in Mimics.

## 2.2. Ligament and load simulation

The mesh models of bones and implants were imported into Abaqus 6.13 (Simulia, Providence, RI, USA) for analysis. The mechanical properties of the ligament and muscle structures were simulated using spring-damping elements, with specific parameter settings detailed in Tables 1 and 2 [12,13,18,19]. To accurately represent the biomechanical behavior of the pelvic ligaments, spring elements were employed to model the anterior sacroiliac ligament, posterior sacroiliac ligament, interosseous sacroiliac ligament, iliolumbar ligament, sacrospinous ligament, sacrotuberous ligament, superior pubic ligament, and arcuate pubic ligament. The origin and insertion points of each ligament were carefully

**Table 1. Lumbar spine material parameters.**

| Material | Elastic modulus, MPa | Poisson ratio | Cross-section area, mm2 |
|---|---|---|---|
| Disc Annulus | 8.4 | 0.45 | |
| Disc Nucleus | Mooney–Rivlin c1 = 0.12, c2 = 0.03 | | |
| Anterior longitudinal ligament | 7 | | 63.7 |
| Posterior longitudinal ligament | 7 | | 20 |
| Ligamentum flavum | 3 | | 40 |
| Intratransverse ligament | 7 | | 1.8 |
| Capsular ligament | 4 | | 30 |
| Interspinous ligament | 6 | | 40 |
| Supraspinous ligament | 6.6 | | 30 |
| Implants | 114000 | 0.3 | |

**Table 2. Pelvic ligament parameters.**

| Material | K, N/m | Number of springs |
|---|---|---|
| Anterior and capsule sacroiliac ligament | 700 | 27 |
| Posterior sacroiliac ligament | 1400 | 15 |
| Interosseous sacroiliac ligament | 2800 | 8 |
| Iliolumbar ligament | 2800 | 30 |
| Sacrospinous ligament | 1400 | 9 |
| Sacrotuberous ligament | 1500 | 15 |
| Superior pubic ligament | 500 | 24 |
| Arcuate pubic ligament | 500 | 24 |

selected based on their anatomical locations. Additionally, material properties, including elastic modulus and Poisson's ratio, were assigned to the intervertebral disc, pubic symphysis, and titanium implants based on data reported in the literature [12,13,18,19]. Material properties for bone, discs, and ligaments, as well as load magnitudes, were adopted from widely cited pelvic FE studies to ensure physiological plausibility and comparability across constructs. Given the study's comparative scope and the small-strain linear-elastic formulation, uniform perturbations of bone elastic modulus are expected to scale absolute responses approximately proportionally, with limited impact on the relative ranking among fixation constructs.

In order to more realistically reflect the constraints in human motion, the pubic symphysis and sacroiliac joints were defined as the constraint regions, while six degrees of freedom were restricted at the bilateral acetabular nodes. A frictional sliding contact was defined between the sacral fracture surfaces, with a friction coefficient of 0.3 [14]. All six degrees of freedom (translational: x/y/z axes; rotational: around x/y/z axes) were restricted at the bilateral acetabular nodes to simulate the physiological fixation of the pelvis to the lower limbs during standing and gait initiation, preventing rigid body motion of the model.A vertical load of 600 N was applied as a distributed load on the superior surface of the L3 vertebra, aiming to replicate the "functional static" loading environment of "transitioning from standing to walking" postoperatively. In biomechanical research, 500 N is commonly used to simulate the upper body weight of an adult in static standing; however, the load transmitted to the pelvis increases to 1.1–1.3 times the static value during the early stance phase of walking. We adopted 600 N (1.2 times 500 N) to balance the basic standing load and the mild dynamic load increment during gait initiation, which is more relevant to clinical real-world scenarios. Full dynamic scenarios were not included in this study and will be addressed in subsequent research.

The generated model is shown in Fig 3 (a schematic diagram), with the constrained region labeled as "bilateral acetabular nodes" and the loaded region labeled as "superior surface of L3 vertebra (distributed 600 N load)".

Four internal fixation models were defined in the study (Fig 4A, 4C, 4E, 4G):

L4L5IS: unilateral L4 pedicle screw + unilateral L5 pedicle screw + iliac screw;

L5S1IS: unilateral L5 pedicle screw + S1 pedicle screw + iliac screw;

L5IS: unilateral L5 pedicle screw + iliac screw;

S1IS: unilateral S1 pedicle screw + iliac screw.

The geometric models of the implants (pedicle screws and iliac screws, both titanium) were created by SolidWorks 2017 (Dassault Systèmes, USA). The core dimensions of the implants were referenced to clinically used specifications: pedicle screws (length: 45 mm, diameter: 6.5 mm) and iliac screws (length: 70 mm, diameter: 7.5 mm). In order to balance computational efficiency and biomechanical accuracy (consistent with previous finite element studies of the pelvis), non-essential structural details (e.g., surface threads) were simplified, while critical dimensions were retained to ensure anatomical fit and mechanical consistency with the actual implants.

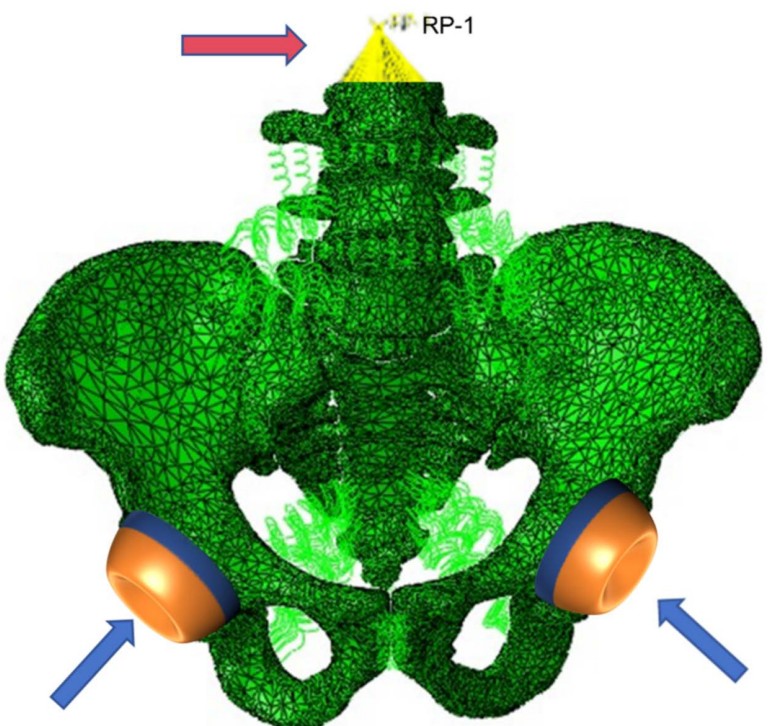

**Fig 3. Pelvic fracture finite element model constructed, Pelvis model mechanics loading (red arrows) and binding constraints (blue arrows).**

After modelling, meshing was performed using an automatic mesh generation method, with the maximum element size of the implant mesh set at 5 mm. The assembled mesh model was then imported into Mimics for material property assignment. All screws were made of titanium alloy material with excellent mechanical properties and biocompatibility.

### 2.3. Finite element model verification and validation

**2.3.1. Model verification (numerical).** We did not perform a formal mesh-quality audit or a dedicated solver-stability study. Instead, we used a standardized mesh of 4-node linear tetrahedra (C3D4; maximum element size 5 mm) and kept all modeling assumptions identical across constructs to ensure fair between-construct comparisons.

**2.3.2. Model validation (experimental/benchmark).** External validity was assessed at the range/trend level against published experiments (Brown/Markolf for spine; Miller for pelvis/SI joint), confirming physiologic ranges and relative stiffness/ordering. Due to the lack of protocol-matched raw datasets, quantitative calibration was not undertaken.

To ascertain the dependability and precision of the model, the spine and pelvis models were validated individually. With respect to the spine component, the vertical displacement experiment conducted by Brown was replicated, and the deformations of the L4 vertebral body in response to forward flexion, backward extension, lateral bending, and torsion motions were calculated and evaluated in comparison with the in vitro experimental data documented by Markolf [20,21].

The pelvic model validation was conducted using the Miller model experimental data and tested with five directions of translational load (294 N) and three directions of rotational load (42 N·m), including translational loads in the anterior-posterior, up-and-down, and internal-and-external directions, as well as torsional loads in flexion, extension, and axial rotation. The test results demonstrated that the response data of the pelvic model exhibited a correlation with the standard error of Miller's experimental data, indicating an overall satisfactory fit [17].

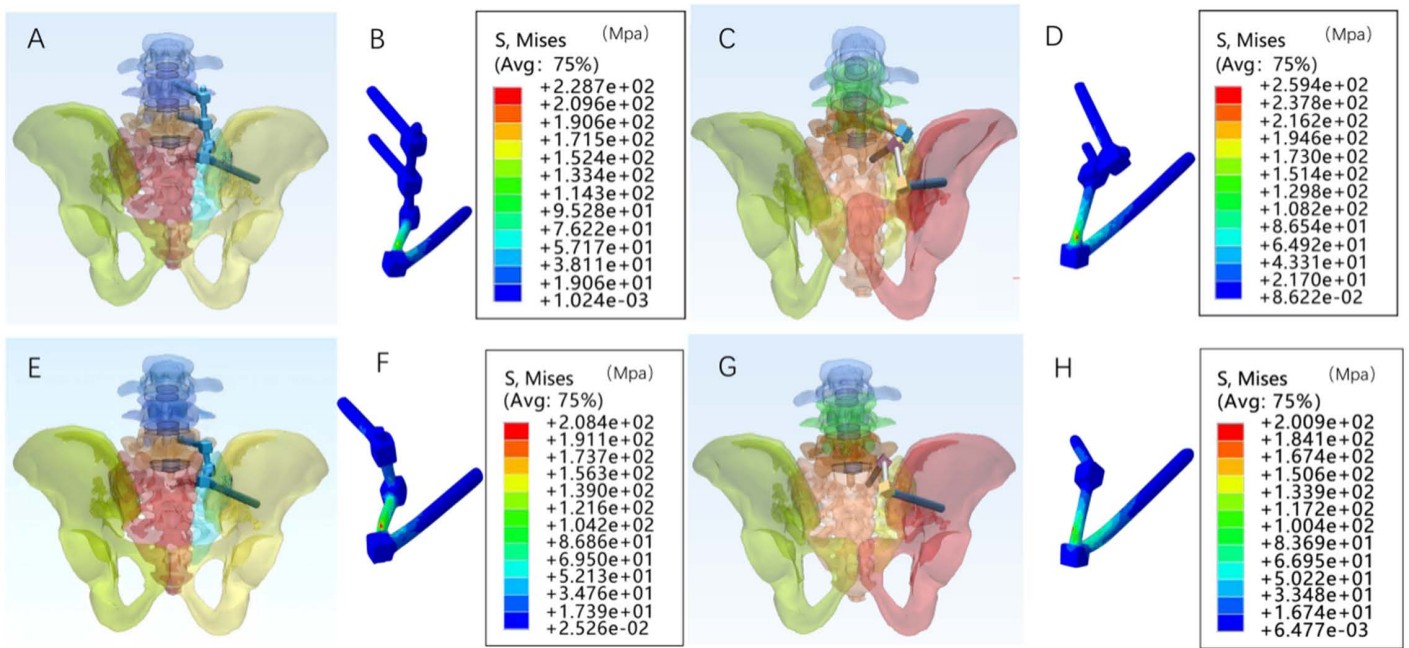

**Fig 4. Schematic diagrams of four iliolumbar fixation models and their corresponding maximum von Mises stress distributions.** A:.unilateral L4 pedicle screw + unilateral L5 pedicle screw + iliac screw. B: Maximum von Mises stress cloud for the L4L5IS model. C:unilateral L5 pedicle screw + S1 pedicle screw + iliac screw. D: Maximum von Mises stress cloud for the L5S1IS model. E:unilateral L5 pedicle screw + iliac screw. F: Maximum von Mises stress cloud for the L5IS model. G: unilateral S1 pedicle screw + iliac screw. H: Maximum von Mises stress cloud for the S1IS model.

### 2.4. Experimental measurements

A vertical force of 600 N was applied to the upper surface of the sacrum to simulate the pressure exerted on the sacrum in a standing position. Boolean operations were performed on the four internal fixation models. With regard to vertical displacement, the normal model was compared with the four aforementioned internal fixation models. Five observation points were taken from the upper surface of the sacrum and used to record the vertical displacements at each of the five points. Point E is the center point of the upper surface of the sacrum.The vertical displacements at each of the five points were then subjected to statistical analysis.

The distribution of stress on the implant components was simulated in a standing position. The maximum von Mises forces on the fixation components were measured, and cloud maps of the stress distribution on the implant structure were obtained.

To assess the displacement of the sacral fracture line, three observation points from proximal to distal were selected on the sacral fracture line, defined as points a, b, and c, respectively. In the simulated standing state a, b, c three points separated the fracture line on both sides of the point to produce a1, a2, b1, b2, c1, c2 six points. These six points were recorded separately in the coordinate system displacement. For each point the X, Y and Z coordinate axis displacements were obtained to record the values. The relative displacement (RD) value was used to indicate the fracture line separation at that observation point. Taking observation point a as an example, the specific formula for separation is as follows:

$$RD = \sqrt{(Xa1 - Xa2)^2 + (Ya1 - Ya2)^2 + (Za1 - Za2)^2}$$

Fracture line displacement was assessed by relative displacement values for different fixation patterns.All quantities are reported in SI units: stress in MPa, force in N, moment in N·m, displacement in mm.

## 3. Results

### 3.1. The distance of vertical displacement of the sacrum

The vertical displacements of five points (A-E) on the upper surface of the sacrum were selected for observation under the action of 600N vertical pressure. The vertical displacements of the above five points of the four groups of models were recorded, and the results are presented in Table 3. The specific results of the vertical displacements of the sacrum in the comparison of the four fixation models are shown in Table 3.

It can be observed that the sacral upper surface vertical displacement ranking (L4L5IS<L5S1IS<S1IS<L5IS) has been measured and the values are presented in Table 3. The objective is to ascertain whether the vertical displacement values are statistically significant. A normality test was conducted on the sacral vertical displacement to ascertain whether the sacral vertical displacement exhibited a normal distribution. The Kolmogorov-Smirnov (V)a and Shapiro-Wilk tests were applied to the sacral vertical displacement, and both normality tests for the four groups of data yielded P>0.05, indicating that the four groups of data conformed to a normal distribution (Table 4).The data from the four groups were subjected to a Levene's test, and the vertical displacement of the sacrum was found to meet the criteria for significance (p>0.05), as detailed in Table 5.The ANOVA test was employed to ascertain whether the sacral displacement exhibited statistically significant differences across the four groups.The ANOVA yielded p<0.001, indicating that the data sets of the four groups were not perfectly equal. This is illustrated in Table 6.A two-by-two comparison LSD test was conducted on the four groups to ascertain whether a relationship existed between the two groups. The LSD test yielded statistically significant results in the comparisons between the groups, with the exception of no statistical difference between L4L5IS and L5S1IS, and no statistical difference between L5IS and S1IS. For further details, please refer to Table 7.

The L4L5IS and L5S1IS models exhibited the most optimal vertical stability of the sacrum among the four groups of models. No statistically significant difference was observed in the values of vertical displacement between these two

**Table 3. Vertical displacement of the upper surface of the sacrum.**

| group | A (mm) | B (mm) | C (mm) | D (mm) | E (mm) | X± S |
|---|---|---|---|---|---|---|
| L4L5IS | 0.6302 | 0.5204 | 0.5374 | 0.6835 | 0.6138 | 0.5971±0.0676 |
| L5S1IS | 0.6380 | 0.5657 | 0.5694 | 0.7157 | 0.6091 | 0.6196±0.0615 |
| L5IS | 0.7411 | 0.6659 | 0.6637 | 0.8044 | 0.7165 | 0.7183±0.0584 |
| S1IS | 0.7397 | 0.6538 | 0.6603 | 0.7719 | 0.6992 | 0.7050±0.0769 |

**Table 4. Sacral vertical displacement normality test.**

| Tests of Normality | | | | | | | |
|---|---|---|---|---|---|---|---|
| | group | Kolmogorov-Smirnovª | | | Shapiro-Wilk | | |
| | | Statistic | df | Sig. | Statistic | df | Sig. |
| Vertical displacement | 1 | .211 | 5 | .200* | .936 | 5 | .635 |
| | 2 | .193 | 5 | .200* | .894 | 5 | .380 |
| | 3 | .215 | 5 | .200* | .913 | 5 | .483 |
| | 4 | .210 | 5 | .200* | .923 | 5 | .549 |

*. This is a lower bound of the true significance.

a. Lilliefors Significance Correction.

Group1: L4L5IS; Group2: L5S1IS; Group3: L5IS; Group4: S1IS.

**Table 5. Sacral vertical displacement Homogeneity of Variances test.**

*Test of Homogeneity of Variances*

| | | Levene Statistic | df1 | df2 | Sig. |
|---|---|---|---|---|---|
| Vertical displacement | Based on Mean | .200 | 3 | 16 | .895 |
| | Based on Median | .097 | 3 | 16 | .960 |
| | Based on Median and with adjusted df | .097 | 3 | 14.588 | .960 |
| | Based on trimmed mean | .205 | 3 | 16 | .891 |

**Table 6. ANOVA test.**

*ANOVA*

*Vertical displacement*

| | Sum of Squares | df | Mean Square | F | Sig. |
|---|---|---|---|---|---|
| Between Groups | .055 | 3 | .018 | 5.120 | .011 |
| Within Groups | .057 | 16 | .004 | | |
| Total | .112 | 19 | | | |

**Table 7. LSD Multiple Comparisons.**

*Multiple Comparisons*

*Dependent Variable: Vertical displacement*

*LSD*

| (I) group | (J) group | Mean Difference (I-J) | Std. Error | Sig. | 95% Confidence Interval | |
|---|---|---|---|---|---|---|
| | | | | | Lower Bound | Upper Bound |
| 1 | 2 | −.0225200 | .0378787 | .560 | −.102819 | .057779 |
| | 3 | −.1212600* | .0378787 | .006 | −.201559 | −.040961 |
| | 4 | −.1079200* | .0378787 | .012 | −.188219 | −.027621 |
| 2 | 1 | .0225200 | .0378787 | .560 | −.057779 | .102819 |
| | 3 | −.0987400* | .0378787 | .019 | −.179039 | −.018441 |
| | 4 | −.0854000* | .0378787 | .039 | −.165699 | −.005101 |
| 3 | 1 | .1212600* | .0378787 | .006 | .040961 | .201559 |
| | 2 | .0987400* | .0378787 | .019 | .018441 | .179039 |
| | 4 | .0133400 | .0378787 | .729 | −.066959 | .093639 |
| 4 | 1 | .1079200* | .0378787 | .012 | .027621 | .188219 |
| | 2 | .0854000* | .0378787 | .039 | .005101 | .165699 |
| | 3 | −.0133400 | .0378787 | .729 | −.093639 | .066959 |

*. The mean difference is significant at the 0.05 level.

Group1: L4L5IS; Group2: L5S1IS; Group3: L5IS; Group4: S1IS.

models. Similarly, no statistically significant difference was noted in the vertical stability of the aforementioned models. The L5IS and S1IS models exhibited a greater vertical displacement than the first two groups of models. The L4L5IS and L5S1IS groups demonstrated superior performance to the L5IS and S1IS groups. It was determined that the L4L5IS and L5S1IS models were of the multisegmental iliolumbar fixation type, comprising two segments of pedicle screw fixation in conjunction with iliac screws. In contrast, the L5IS and S1IS models were of the one-segment fixation type, combining a single segment of pedicle screw fixation with iliac screws. The combination of two segments of pedicle screw fixation demonstrated superior stability compared to the one-segment fixation type.

No statistically significant difference was observed between the two-segment fixation and the two-segment combination with respect to the vertical stability of the sacrum. Similarly, no statistical difference was noted between single-segment fixation and single-segment vertical stability, with the latter exhibiting comparable results.

### 3.2. The maximum von Mises stress in the implant

The maximum von Mises stresses in the L4L5IS, L5S1IS, L5IS, and S1IS groups under a 600 N vertical load in the Denis II fracture model are 228.7 MPa, 259.4 MPa, 208.4 MPa,200.9 MPa(Fig 4B, 4D, 4F, 4H). The maximum von Mises stress in L5S1IS is the largest among the four groups of models, followed by the L4L5IS maximum von Mises stress of 228.7 MPa. The remaining two groups of models exhibit similar maximum von Mises stress values. An analysis of the internal fixation distribution cloud diagram revealed that the maximum von Mises stress in the two-segment pedicle screw combined with iliac bone screw fixation model was greater than that in the single-segment pedicle screw combined with iliac bone screw fixation model. This indicates that the risk of internal fixation fracture is elevated in the two-segment pedicle screw and iliac bone screw fixation model. It should be noted that the maximum von Mises stress values of all models were below the yield stress of titanium (1050 MPa) [22]. The maximum von Mises stress around the screws in all models was as follows: L4L5IS: 33.48 MPa > L5IS: 20.58 MPa > L5S1IS: 15.77 MPa > S1IS: 15.52 MPa. In all models, the stress was lower than the cortical bone yield stress of 150 MPa [23].

A review of the implant stress distribution cloud diagram (Fig 4B, 4D, 4F, 4H) revealed that the stress in the implant was concentrated at the iliac screw nail tail and the iliac screw connector. These two locations were identified as the most susceptible to implant loosening and fracture, particularly in areas where the implant screw stress was concentrated.

### 3.3. Relative displacement of the fracture line

Under the vertical force of 600 N, the relative displacement values of the observation points of a, b and c fracture lines on the longitudinal fracture line of the sacrum of the four groups of models were counted, and the specific data are shown in Fig 5. The above data can be seen that the data results are close to those of each group. a, b, and c observation points, the closer to the proximal end of the relative displacement value is larger, and the closer to the distal end of the relative displacement value is smaller.

## 4. Discussions

Sacral fractures are severe pelvic injuries with rising incidence due to high-energy trauma (e.g., motor vehicle accidents, falls) [24–26]. They impair mobility and cause chronic complications (pain, incontinence), severely affecting patients' quality of life [27–29], highlighting the need for timely effective treatment.

Current primary approaches for longitudinal sacral fractures include: (1) Sacroiliac screw fixation: The gold standard for its minimally invasive nature and pelvic stability restoration, achieving effective immobilization with reduced complications via precise reduction. However, it is limited by intact anatomical structures and adequate bone density; poor preoperative reduction raises risks of intraoperative neurovascular injury [30]. (2) Triangular osteosynthesis: Combines sacroiliac screws with iliolumbar fixation, enhancing stability through synergistic transverse-longitudinal stabilization. Despite higher surgical complexity, it enables earlier functional rehabilitation in specific cases via superior postoperative stability [4,5,7,31]. (3) Iliolumbar fixation: Matches the aforementioned techniques in stability. Minimally invasive procedures are now well-established; Jazini et al. [32] validated iliolumbar fixation support stability in cadaveric models, with cement-augmented pedicle screws further enhancing stability. It is advantageous for patients with malreduction or neurological injuries (allowing simultaneous reduction and neural exploration) but carries higher risks of surgical trauma, infection, hardware protrusion, and screw fracture [33].

 

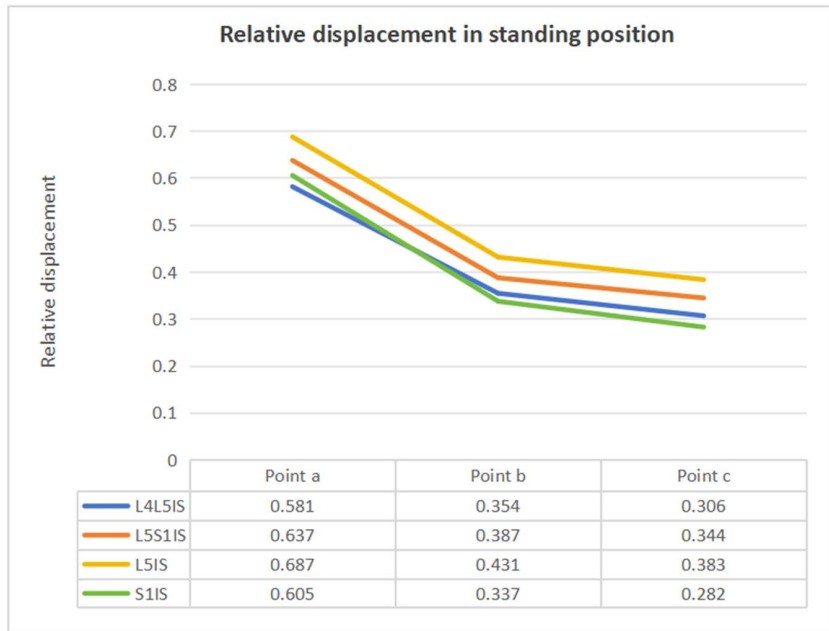

**Fig 5. Relative displacement values for each model group.**

While clinical studies confirm minimally invasive iliolumbar fixation efficacy [34,35], Iliolumbar fixation remains an important technique in clinical care especially when sacroiliac screws are not available.

Rather than restating numerical outputs, we interpret the findings for iliolumbar segment selection in unilateral longitudinal sacral fractures. Under identical boundary conditions, dual-segment constructs (L4+L5+IS; L5+S1+IS) reduced sacral vertical displacement compared with single-segment constructs ($p < 0.05$ where indicated), while all peak implant stresses remained below the titanium alloy yield strength. These patterns frame a clinical trade-off between greater stability and preservation of lumbar motion.

Among the four models, L5+S1+IS achieved stability comparable to L4+L5+IS (no significant difference), while avoiding proximal extension and additional lumbar burden. Accordingly, for unilateral longitudinal sacral fractures (AO C1.3, Denis II), L5+S1+IS may be prioritized when S1 pedicle integrity permits. When S1 pedicle integrity is compromised, L4+L5+IS provides comparable stability ($p > 0.05$ vs L5S1IS) and can be selected as an alternative. Short-segment options (S1IS/L5IS) remain useful for motion preservation; however, their greater displacement than dual-segment constructs warrants graduated weight-bearing and core-stabilization rehabilitation to mitigate secondary displacement during healing.

Dual-segment models exhibited higher peak von Mises stresses (L5S1IS: 259.4 MPa; L4L5IS: 228.7 MPa) than single-segment groups (L5IS: 208.4 MPa; S1IS: 200.9 MPa), yet all values were well below the titanium yield strength (1050 MPa) [22]. Bone–screw interface peaks (33.48 MPa) were below cortical yield thresholds (150 MPa) [23]. Consistent stress localization at the iliac screw tail and connector delineates a plausible interface-related risk zone; early hardware removal after union may be considered case by case.

Under a 600 N vertical load, proximal fracture-line displacements (points a, b, c) exceeded distal measurements (Fig 5), in keeping with lumbopelvic load-transfer patterns. The relatively higher proximal demands may help explain clinically observed delayed union in this region [30]. Notably, unilateral iliolumbar fixation primarily re-routes load transfer and does not directly span the fracture; this is consistent with its supportive rather than anatomic fixation role when sacroiliac screws cannot be used.

Our static, small-strain analysis does not capture fatigue under cyclic activities (walking, stairs), where sub-yield stresses at the iliac screw connector could accumulate damage. Implant loosening risk may rise at high-stress interfaces and in patients with low bone mineral density (BMD) or unfavorable sacral morphology. When short-segment constructs are chosen to preserve motion, we recommend phased weight-bearing, core stabilization, and vigilant follow-up for early signs of loosening. Future finite element work should incorporate cyclic loading and BMD-informed material mapping to quantify fatigue safety margins.

Prior FE research has mainly centered on sacroiliac screws and triangular/combined constructs [4,5,11,14], with comparatively fewer analyses isolating the effect of iliolumbar segment selection itself. Within the iliolumbar framework, our results clarify that upgrading from single- to dual-segment iliolumbar fixation reduces sacral displacement, and that L5+S1+IS and L4+L5+IS achieve comparable stability while implying different motion-preservation trade-offs. The observed connector-level stress concentration complements SI-screw–centric studies that mostly varied screw length/trajectory on the sacral side [11,14], adding iliolumbar-specific guidance for segment/level choice.

Several limitations should be noted. First, to isolate posterior-ring mechanics we preserved the anterior ring and simplified soft-tissue modeling (key ligaments retained, but no muscles, detailed joint compliance, or muscle–joint interactions); fracture-line displacement was used descriptively without formal statistical testing. Second, we did not incorporate patient-specific parameters (BMD distribution, sacral morphology, age) or dynamic/cyclic loading, which may affect absolute displacement/stress and the appraisal of fatigue-related loosening, limiting clinical extrapolation to anatomies and bone quality similar to our assumptions. Third, we did not perform mesh-convergence, numerical verification (mesh-quality metrics/solver stability), sensitivity analysis, or protocol-matched quantitative validation (correlation/RMSE). Taken together, our findings are best interpreted as relative, between-construct comparisons under uniform assumptions. Future work will include subject-specific FE with BMD-mapped properties, cyclic-fatigue simulations, improved frictional/cohesive implant–bone interfaces, and parametric convergence/sensitivity/validation studies.

## 5. Conclusion

For unilateral vertical sacral fracture (AO C1.3 Denis II), Based on these biomechanical insights, we propose a decision-making framework that tailors fixation strategy to individual patient characteristics: (1) For patients with intact S1 pedicles and high stability demands, dual-segment constructs (L5+S1+iliac screws) provide superior mechanical performance (demonstrating the minimal vertical displacement among all groups, $p < 0.05$); (2) When S1 anatomy is compromised, L4+L5+iliac fixation offers equivalent stability (no significant difference vs L5S1IS, $p > 0.05$); (3) In mobility-sensitive cases, short-segment configurations (S1/L5+iliac screws) achieve functional rehabilitation goals despite requiring supplemental postoperative stabilization (significantly greater displacement vs dual-segment).

## Supporting information

**S1 Fig. L4L5IS fixation mode: Displacements of 5 points on the upper surface of the sacrum.** This figure displays the displacement data of 5 pre-designated measurement points on the upper surface of the sacrum under the L4L5IS fixation mode. Displacement values are presented in three spatial directions (X: anteroposterior, Y: mediolateral, Z: craniocaudal) and are derived from finite element analysis testing.
(TIF)

**S2 Fig. L5S1IS fixation mode: Displacements of 5 points on the upper surface of the sacrum.** This figure displays the displacement data of 5 pre-designated measurement points on the upper surface of the sacrum under the L4L5IS fixation mode. Displacement values are presented in three spatial directions (X: anteroposterior, Y: mediolateral, Z: craniocaudal) and are derived from finite element analysis testing.
(TIF)

**S3 Fig. L5IS fixation mode: Displacements of 5 points on the upper surface of the sacrum.** This figure displays the displacement data of 5 pre-designated measurement points on the upper surface of the sacrum under the L4L5IS fixation mode. Displacement values are presented in three spatial directions (X: anteroposterior, Y: mediolateral, Z: craniocaudal) and are derived from finite element analysis testing.
(TIF)

**S4 Fig. S1IS fixation mode: Displacements of 5 points on the upper surface of the sacrum.** This figure displays the displacement data of 5 pre-designated measurement points on the upper surface of the sacrum under the L4L5IS fixation mode. Displacement values are presented in three spatial directions (X: anteroposterior, Y: mediolateral, Z: craniocaudal) and are derived from finite element analysis testing.
(TIF)

## Author contributions

**Data curation:** Yupeng Ma, Weiwei Liu, Tao Huang, Huanyu Hong, Yong Zhao, Guofeng Xu.

**Formal analysis:** Guofeng Xu.

**Funding acquisition:** Yong Zhao.

**Writing – original draft:** Yupeng Ma, Yu Li.

**Writing – review & editing:** Guofeng Xu, Yu Li.

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
