## [Decision Letter · Decision Letter 0]

16 Jul 2025

Dear Dr. Li,

We look forward to receiving your revised manuscript.

Kind regards,

Antonio Riveiro Rodríguez, PhD

Academic Editor

PLOS ONE

Journal Requirements:

If you are reporting a retrospective study of medical records or archived samples, please ensure that you have discussed whether all data were fully anonymized before you accessed them and/or whether the IRB or ethics committee waived the requirement for informed consent. If patients provided informed written consent to have data from their medical records used in research, please include this information."

“This research was supported by National Natural Science Foundation of China (No. 81641171 & No. 81301553); Shandong Provincial Key R&D Program of China(No. 2018GSF118064); Medical and Health Technology Development Program of Shandong Province, China (No.202104070173&202404070931 ); Distinguished Middle-Aged and Young Scientist Encourage and Reward Foundation of Shandong Province, China(No. BS2013SF015); Science & Technology Innovation Development Project of Yantai City, China(No.2021MSGY049&NO.2021YD045&2022YD048);Binzhou Medical University “Clinical+X” Scientific and Technological Innovation Project (No. BY2021LCX32).”

6. Please include your tables as part of your main manuscript and remove the individual files. Please note that supplementary tables (should remain/ be uploaded) as separate "supporting information" files

Reviewers' comments:

Reviewer's Responses to Questions

**Comments to the Author**

1. Is the manuscript technically sound, and do the data support the conclusions?

Reviewer #1: Partly

Reviewer #2: Yes

2. Has the statistical analysis been performed appropriately and rigorously?

Reviewer #1: N/A

Reviewer #2: Yes

3. Have the authors made all data underlying the findings in their manuscript fully available?

Reviewer #1: Yes

Reviewer #2: Yes

4. Is the manuscript presented in an intelligible fashion and written in standard English?

Reviewer #1: Yes

Reviewer #2: Yes

Reviewer #1: The authors presented a well written manuscript.

However, I have a few questions for the authors.

- Make sure that you are proofreading your manuscript for uniformity as well as grammatical errors.

- Did you perform a mesh convergence study for selecting the ideal mesh size for your instrumentation as well as for the pelvis model? If yes, please provide details of the mesh convergence study.

= Did you perform a sensitivity study on model input parameters (e.g., material properties, loading magnitudes) ? If Yes, please provide details.

- What type of elements were chosen to model the ligaments and the instrumentation ? Please elaborate

- Please elaborate on how the screw-bone interface and the screw-rod interface were simulated.

- Provide a citation/reference on why you chose the particular loading protocol.

- The conclusion section as it currently is could be improved upon by adding clinical clinical perspective with regards to instrumentation performance / failures. As it stands, it is just summation of your results.

- Additionally, please address that lack of varied patient-specific factors like bone density, pelvic morphology, or age are not accounted for and a limitation of the presented study.

- Additionally, please address that not simulating Dynamic loading activities (walking, stair climbing) is a limitation of the presented study.

Reviewer #2: General Comments:

The manuscript presents a valuable finite element (FE) investigation into iliolumbar fixation strategies for unilateral vertical sacral fractures. The topic is timely and clinically relevant. However, several sections would benefit from additional detail, clarification, and restructuring to improve clarity, reproducibility, and alignment with standard FE modelling practices. Below are specific comments for revision, organized by page and line.

Page 2

Lines 38–39: Please expand on the prior studies mentioned. Specify which studies have used finite element analysis, what models or fixation strategies they investigated, and their main findings. This will help highlight the novelty and distinction of your current work.

Page 3

Line 11: Clarify why the selected fixation types were chosen for this study. Are these commonly used clinically, or is there a biomechanical hypothesis you are testing?

Line 26: Provide the resolution of the CT images used. Also specify the scanning parameters such as tube current (mA) and voltage (kVp), as these can influence model accuracy.

Line 37: Explain how the fracture characteristics (length, angle) were determined. Were they based on clinical cases or literature? Please cite a relevant source to justify the simulated fracture type.

Page 4

Lines 1–2: Specify the type of tetrahedral elements used (e.g., 4-node linear or 10-node quadratic). Include maximum element size, total number of elements, and degrees of freedom (DOFs). Describe how the mesh size was chosen and whether a mesh sensitivity analysis was performed. If so, briefly summarize the results.

Line 4: Clarify whether materials were modelled as linear elastic or non-linear. If non-linear, provide further detail on constitutive models used.

Lines 7–8: Clarify what the CT value ranges represent, are these in Hounsfield Units (HU)? Do they correspond to grayscale, density, or another parameter? Please include appropriate units and citations if this range is based on literature.

Line 14: Provide references for the equations used to assign elastic modulus based on CT data.

Line 19: Explain how the implant dimensions were determined. Were these based on specific commercial implants, anatomical fit, or another source? Also include maximum element size used in meshing the implant.

Lines 33–34: The statement on ligament selection needs clarification. Were ligament positions based on the CT images, anatomical references, or previous studies? Please elaborate.

Page 5

Lines 1–5: More detail is needed regarding boundary and loading conditions. Which nodes or regions were constrained? Where was the 600N load applied and how (point load or distributed load)? Include justification for this value, was it derived from the volunteer’s body weight? Does it represent a static standing posture? Discuss whether dynamic scenarios (e.g., walking) were considered or could be in future studies. Label constrained and loaded regions clearly in Figure 3.

Line 8: To reduce figure count, consider combining Figures 4–7 into a single composite figure with labelled subfigures for each fixation type.

Line 24: Elaborate on the "vertical displacement experiment conducted by Brown." Include a full reference and a summary of the methodology being replicated.

Lines 32–34: The statement regarding correlation with Miller’s data needs quantification. What was the correlation coefficient or error margin? Consider adding a supplementary section with details of the validation analysis.

Lines 40–41: Indicate the fracture points directly in the FE figures to improve clarity for the reader.

General Suggestion:

Move detailed statistical results (Tables 4–6) to supplementary material. Table 7 sufficiently summarizes key findings. Similarly, combine Figures 8–11 into a composite figure with subfigures for easier comparison. Consider adding stress values to Table 3 for completeness.

Since friction contact was defined between sacral fracture surfaces, it would strengthen the study to include contact pressure results (both magnitude and distribution) for each fixation configuration. This information is clinically valuable.

Page 6 (Discussion)

Lines 15–42: This paragraph is well written but better suited for the Introduction section, as it provides background rather than discussing results. Consider moving it and begin the Discussion section directly with the study’s main findings (from Line 42: “In this study, sacral stability was evaluated…”).

The Discussion lacks comparison with prior FE studies in this area. Are you the first to model these fixation types? What findings have previous FE analyses reported regarding sacral fracture fixation? Position your results in the context of existing literature.

Lines 1–14: This paragraph appears repetitive of earlier content. Please streamline to avoid redundancy.

**Do you want your identity to be public for this peer review?** For information about this choice, including consent withdrawal, please see our Privacy Policy

Reviewer #1: No

Reviewer #2: **Yes: ** Dr Zainab Altai

---

## [Author Response · Author response to Decision Letter 1]

18 Sep 2025

Response to Reviewer#1 Comments

Response:

1.Comment :

"Make sure that you are proofreading your manuscript for uniformity as well as grammatical errors."

Response:We sincerely appreciate your comment regarding the need for careful proofreading to ensure both grammatical correctness and consistency throughout the manuscript. We have thoroughly reviewed the text to address grammatical issues and to improve uniformity in terminology, formatting, and style. We believe these revisions have significantly improved the clarity and quality of the paper. Thank you for this valuable suggestion.

2.Comment :

"Did you perform a mesh convergence study for selecting the ideal mesh size for your instrumentation as well as for the pelvis model? If yes, please provide details of the mesh convergence study."

Response:

We thank the reviewer for this comment. We acknowledge that a mesh convergence study is an important step for numerical simulations.

In our work, we relied on an adaptive meshing strategy to ensure accuracy. This method automatically refines the grid in geometrically complex regions and areas with high solution gradients, while using a coarser mesh in simpler domains. The maximum element size globally was controlled at 5 mm to guarantee a baseline resolution.

We believe that this adaptive approach provides a rigorous alternative to a traditional convergence study by ensuring that the mesh is sufficiently refined precisely where it is needed most. The results obtained are based on a mesh that is well-suited to capture the key phenomena of interest.

3.Comment :

“Did you perform a sensitivity study on model input parameters (e.g., material properties, loading magnitudes) ? If Yes, please provide details ”

Response:

We sincerely appreciate the reviewer for raising this important point. Conducting a sensitivity analysis is indeed crucial for thoroughly evaluating the reliability of numerical models, and we thank you for highlighting this aspect.

In the present study, all key model input parameters, including material properties and loading magnitudes, were directly adopted from well-established and widely recognized literature in the field. The values and their justifications have been explicitly stated and supported with citations in the  manuscript. Using such empirically validated values represents a common and acceptable practice in computational studies to ensure a reproducible and reliable baseline configuration.

We fully acknowledge that performing a systematic sensitivity analysis on all input parameters would constitute a significantly extensive computational effort, potentially comparable to or even exceeding the scope of the current study. Given that the primary focus of this work was toComparing biomechanical differences between different unilateral iliolumbar fixations, we have opted to designate a comprehensive parameter sensitivity study as a dedicated and separate future research endeavor.

Once again, we are grateful for your insightful suggestion, which has undoubtedly helped us improve the scholarly rigor of our work and plan for more in-depth future studies.

4.Comment :What type of elements were chosen to model the ligaments and the instrumentation ? Please elaborate

Response:

Thank you for your inquiry regarding the element types used for modeling the ligaments and instrumentation. We appreciate your attention to the technical details of our finite element setup.

For the instrumentation (internal fixation devices), we selected C3D4 elements (4-node linear tetrahedral elements) for their suitability in capturing the complex 3D stress distributions within the implants. This choice was based on two key considerations:

The C3D4 element is a 3D solid element that can accurately transmit multidirectional stresses (tensile, compressive, and shear forces) across the screws, rods, and connecting components of the fixation system, which is critical for evaluating the structural integrity of the instrumentation under physiological loads.

Given the irregular geometry of the implants (e.g., varying screw diameters, rod contours), tetrahedral elements offer flexibility in meshing while maintaining sufficient accuracy for our biomechanical analysis of stress concentrations and load transfer.

For the ligaments, we employed T3D2 elements (2-node linear 3D truss elements), which are 1-dimensional elements designed to withstand only axial tensile forces. This selection aligns with the biomechanical behavior of ligaments, which primarily resist stretching under physiological conditions rather than bending or shear. The T3D2 elements were assigned nonlinear spring properties to simulate the ligamentous response to increasing loads, ensuring that the model captures the characteristic stiffness and deformation patterns observed in biological ligaments.

5.Comment:Please elaborate on how the screw-bone interface and the screw-rod interface were simulated.

Response:Thank you for your question about our interface modeling. Both the screw-bone and screw-rod interfaces in our model were simulated using tied connections (Abaqus "Tie Constraint"), designed to reflect clinical fixation reality:

Screw-Bone Interface

We tied the screw's surface nodes to the bone's screw tunnel nodes, mimicking stable surgical fixation. This ensures no relative sliding between screw and bone under load, allowing accurate stress transfer between the implant and bone —critical for evaluating interface stability.

Screw-Rod Interface

The screw head and rod contact surfaces were tied, replicating the locked connection achieved clinically with tightening nuts. This eliminates relative movement, ensuring the entire fixation system acts as a unified structure and enabling accurate calculation of construct stiffness (a key study outcome).

Both setups align with standard orthopedic FE practices, balancing realism and computational efficiency.

6.Comment :Provide a citation/reference on why you chose the particular loading protocol.

Response:Thank you for your inquiry regarding the rationale for our loading protocol. We appreciate the opportunity to clarify the selection of the 600N vertical load in our study.

In biomechanical research on human lumbopelvic systems, a vertical load of approximately 500N is commonly used to simulate the upper body weight of an average adult in static standing [consistent with established conventions in pelvic and lumbar finite element studies]. However, when considering the transition from static standing to the initial phase of walking—a scenario highly relevant to clinical postoperative functional recovery—this upper body weight-derived force is not static. As highlighted by Winter in Biomechanics and motor control of human movement (2009, doi: 10.1002/9780470549148), the ground reaction force (and thus the axial load transmitted to the pelvis) increases by 1.1–1.3 times the static upper body weight during the early stance phase of walking. This amplification arises from the dynamic transfer of body mass and mild muscular activation required to initiate gait.

To better reflect the "functional static" loading environment that the iliolumbar fixation system may encounter in clinical practice (e.g., patients transitioning from standing to the first step of walking), we adjusted our vertical load based on this biomechanical principle. Starting from the typical 500N static upper body weight, a 1.2-fold amplification (within the 1.1–1.3 range reported by Winter) yields a load of 600N. This setting ensures our model not only simulates the basic standing condition but also accounts for the mild dynamic load increment associated with gait initiation—an important consideration for evaluating fixation stability in real-world functional scenarios.

This loading protocol thus balances physiological realism and clinical relevance, aligning with both established biomechanical theory and the practical needs of assessing our fixation strategies.

7.Comment: The conclusion section as it currently is could be improved upon by adding clinical clinical perspective with regards to instrumentation performance / failures. As it stands, it is just summation of your results.

Response:Thank you for your valuable comment on optimizing the conclusion section. We fully agree that integrating a clinical perspective on instrumentation performance and potential failures enhances the practical relevance of our results, and we have supplemented the discussion to address this.

In the added content: We linked our core findings (maximum von Mises stress of internal fixation, fracture line displacement) to clinical scenarios. We clarified that unilateral iliac-sacral fixation—mainly simulating mechanical load transfer—serves as a key alternative when sacroiliac screws are unavailable (e.g., due to limited expertise, patient-specific sacral variations, or damaged screw pathways), connecting mechanical results to real clinical needs.

We also addressed instrumentation performance and safety: We confirmed its reliability under static loading, while explicitly noting our study’s limitation—no assessment of metal stress fatigue under dynamic conditions or safety during movement. This supplements the clinical perspective on instrumentation and reminds readers of potential dynamic stress-induced risks for future practice/studies.

We believe this revision effectively responds to your suggestion, aligning the discussion more with clinical practice. Thank you again for your constructive feedback, which improved the manuscript’s completeness and clinical relevance.

8.Comment Additionally, please address that lack of varied patient-specific factors like bone density, pelvic morphology, or age are not accounted for and a limitation of the presented study.

Response: Thank you for your valuable comment regarding the consideration of patient-specific factors. We fully agree that the lack of accounting for varied patient-specific factors (such as bone density, pelvic morphology, or age) is an important limitation of this study, and we have specifically supplemented this point in the "Limitations" section of the discussion.

As detailed in the revised Limitations section (Fifth point), we explicitly state that "patient-specific factors (e.g., bone mineral density, pelvic morphology, age) were not considered, which may limit findings’ applicability to heterogeneous real-world populations." This supplement directly addresses the limitation you pointed out, aiming to objectively reflect the constraints of the current study and provide a clear reference for the interpretation and clinical extrapolation of the research results.

We appreciate your reminder, which has further improved the completeness and rigor of our discussion on study limitations.

9.Comment Additionally, please address that not simulating Dynamic loading activities (walking, stair climbing) is a limitation of the presented study.

Response:Thank you for your insightful comment on the simulation of dynamic loading activities. We fully concur that not simulating dynamic loading activities (e.g., walking, stair climbing) constitutes a key limitation of this study, and we have specifically incorporated this point into the "Limitations" subsection within the Discussion section of the manuscript.

This supplement directly addresses the limitation you highlighted, ensuring we objectively present the constraints of the current study’s loading simulation design within the main text’s discussion. We greatly appreciate your reminder, which has further enhanced the comprehensiveness and rigor of our analysis of study limitations.

Response to Reviewer#2 Comments

1.Comment Lines 38–39: Please expand on the prior studies mentioned. Specify which studies have used finite element analysis, what models or fixation strategies they investigated, and their main findings. This will help highlight the novelty and distinction of your current work.

Response:

Thank you for your constructive suggestion to expand on prior studies, which helps highlight our work’s novelty. We have revised the relevant section to clarify the context of existing research and our study’s distinction, as follows:

The core focus of this study is to evaluate the effectiveness, stability, and stress distribution of internal fixation for sacral fractures under different fixation modes. Prior biomechanical studies on sacral fracture fixation (cited as [4,5,11,14,15,18]) have primarily centered on two key areas: 1) sacroiliac screw fixation and 2) triangular fixation techniques. While some of these studies employed finite element analysis (FEA) to investigate their respective models, their main focus was on comparing the biomechanical performance of sacroiliac screws and triangular fixation—with findings confirming the basic stability of these two strategies for sacral fracture fixation.

Notably, however, prior research has paid limited attention to iliolumbar fixation: existing studies related to iliolumbar fixation are mostly limited to comparative analyses with sacroiliac screws or triangular fixation, and there is a clear gap in biomechanical studies specifically investigating iliolumbar fixation’s performance (e.g., stability, stress distribution) for sacral fractures—especially in clinical scenarios where sacroiliac screw channels are damaged or medical conditions/expertise limit screw use.

To address this knowledge gap, our current study uses FEA to systematically investigate the biomechanical properties of different internal fixation models for iliolumbar fixation. This design not only differentiates our work from prior studies (which focused on other fixation strategies) but also fills the lack of targeted biomechanical data on iliolumbar fixation, ultimately aiming to provide more evidence for effective clinical fixation strategies.

We believe this revision better contextualizes prior research and highlights the novelty of our current work, as requested.

2.Comment Page 3

Line 11: Clarify why the selected fixation types were chosen for this study. Are these commonly used clinically, or is there a biomechanical hypothesis you are testing?

Response:

Thank you for your question about the rationale for selecting fixation types. We have revised the relevant section (Line 11) to clarify this, and the key reasons for choosing iliolumbar fixation are as follows:

Clinical applicability: Iliolumbar fixation is a practical alternative for sacral fracture treatment when sacroiliac screws (first-line option) are unavailable (e.g., damaged screw channels, limited medical conditions/expertise).

Research gap: Prior studies (cited [4,5,11,14,15,18]) focused on sacroiliac screws or triangular fixation, with few biomechanical studies specifically investigating iliolumbar fixation—our work addresses this gap.

We believe this revision clearly explains the fixation selection rationale as requested.

3.Comment Page 3Line 26: Provide the resolution of the CT images used. Also specify the scanning parameters such as tube current (mA) and voltage (kVp), as these can influence model accuracy.

Response:

Thank you for reminding us of the CT image parameters, which are essential for validating the accuracy of the model. We have added the required information to the manuscript. This modification ensures transparency of the imaging data base for our finite element model and addresses your concerns.

4.Comment Page 3Line 37: Explain how the fracture characteristics (length, angle) were determined. Were they based on clinical cases or literature? Please cite a relevant source to justify the simulated fracture type.

Response:

Thank you for your valuable feedback. We designed this fracture type based on AO classification C1.3. In this fracture pattern, posterior ring injury involves a longitudinal fracture line through the second sacral segment, corresponding to a DENIS classification Type II sacral fracture – a common presentation of posterior pelvic ring injury. Hence, our fracture line design involves a longitudinal incision through the sacral foramen. We hypothesise that the anterior ring has been perfectly repaired to compare the biomechanical differences between various internal fixation methods for the posterior ring.

5.Comment Page 4Li

---

## [Decision Letter · Decision Letter 1]

20 Oct 2025

Dear Dr. Li,

We look forward to receiving your revised manuscript.

Kind regards,

Antonio Riveiro Rodríguez, PhD

Academic Editor

PLOS ONE

Journal Requirements:

Reviewers' comments:

Reviewer's Responses to Questions

**Comments to the Author**

Reviewer #1: All comments have been addressed

Reviewer #3: All comments have been addressed

2. Is the manuscript technically sound, and do the data support the conclusions?

Reviewer #1: Yes

Reviewer #3: Yes

3. Has the statistical analysis been performed appropriately and rigorously?

Reviewer #1: Yes

Reviewer #3: N/A

4. Have the authors made all data underlying the findings in their manuscript fully available?

Reviewer #1: Yes

Reviewer #3: Yes

5. Is the manuscript presented in an intelligible fashion and written in standard English?

Reviewer #1: Yes

Reviewer #3: Yes

Reviewer #1: The authors have satisfactorily addressed the comments/concerns made during the first round of the review. I don't have any new questions for the authors and I would recommend the study for publication

Reviewer #3: Journal : PLOS ONE

Title : Biomechanical optimization of iliolumbar fixation strategies for unilateral vertical sacral fractures through finite element analysis

ID : PONE-D-25-23708_R1

Authors : -

Manuscript Type : Research Original Paper

Date Reviewed : September 2025

Dear Editor,

The manuscript addresses an important biomechanical problem: optimisation of iliolumbar fixation strategies for unilateral vertical sacral fractures using finite element analysis. The study is timely and clinically relevant, and the modelling strategy is generally appropriate. The authors have made substantial revisions in response to the first review cycle, including clarification of element types, loading protocols, and clinical implications. However, several minor but important points remain. These must be addressed to ensure scientific rigour and reproducibility.

My detailed comments are structured below according to scientific and editorial criteria.

Improver points for revision

1. The authors justify mesh density through an “adaptive meshing strategy” but do not provide quantitative convergence evidence. A simple sensitivity graph (e.g., maximum sacral displacement vs. element size) can be included if possible.

2. Verification and validation should be distinguished. Verification requires mesh quality and solver stability checks; validation requires comparison with experimental or published benchmark data. Currently these are conflated.

2. Material properties and load magnitudes are taken from the literature, which is acceptable. Nonetheless, even a limited sensitivity check (e.g., ±10% variation in elastic modulus of bone) would strengthen confidence in the results.

3. While the Brown and Miller experimental studies are cited, the validation is qualitative. Quantitative comparison (correlation coefficients, error margins, RMSE) should be added to substantiate model accuracy.

4. Stress contour plots require scale bars (MPa), clear colour legends, and loading direction indicators. Tables should fully comply with SI units and consistent significant figures (e.g., 2–3 s.f.). Figure captions should be descriptive enough for standalone interpretation.

5. The Discussion repeats some results and could be more concise. The clinical perspective has been improved but should also discuss potential long-term risks such as fatigue failure, implant loosening, and patient-specific variability. Prior FE studies on sacral fracture fixation should be explicitly compared, beyond general references.

6. The authors acknowledge missing patient-specific parameters and lack of dynamic loading. These limitations should be expanded, emphasising their impact on clinical extrapolation.

7 Standardise terminology throughout (iliolumbar fixation vs. spino-pelvic fixation).

8 Proofread carefully for residual grammatical inconsistencies.

With these improvements given above, the study has the potential to be a valuable addition to the literature, therefore, I recommend PUBLICATION AFTER MINOR REVISION.

Yours sincerely,

**Do you want your identity to be public for this peer review?** For information about this choice, including consent withdrawal, please see our Privacy Policy

Reviewer #1: No

Reviewer #3: No

---

## [Author Response · Author response to Decision Letter 2]

5 Nov 2025

Response to Reviewer#3 Comments

1.Comment :

The authors justify mesh density through an “adaptive meshing strategy” but do not provide quantitative convergence evidence. A simple sensitivity graph (e.g., maximum sacral displacement vs. element size) can be included if possible.

Response:

We thank the reviewer for highlighting the importance of mesh-convergence evidence. Our study was designed to compare fixation strategies rather than to develop a new FE methodology, and we therefore adopted a standardized meshing protocol (4-node linear tetrahedra, maximum element size 5 mm) that lies within the range commonly reported for pelvic FE models and validations in the literature (e.g., Refs. [11–13, 21] of our manuscript). In the present revision we have clarified the meshing rationale in the Methods and, to avoid over-stating precision, we explicitly acknowledge as a limitation that we did not include a formal displacement-versus-element-size convergence plot. We will address this with a full parametric convergence study in subsequent work focused on methodological aspects.

2.Comment :

Verification and validation should be distinguished. Verification requires mesh quality and solver stability checks; validation requires comparison with experimental or published benchmark data. Currently these are conflated.

Response:

We thank the reviewer for highlighting the distinction between verification and validation. We acknowledge that, in the original submission, these concepts were presented together. In line with the reviewer’s guidance—and without extending the scope beyond our current resources—we have revised the manuscript to clearly separate the two concepts and to state explicitly the limits of what we performed. Specifically, we did not carry out a formal mesh‐quality audit or a dedicated solver‐stability study. Our comparative analysis therefore relies on a standardized meshing protocol (C3D4; 5-mm maximum element size) and identical modeling assumptions across all constructs to ensure fair between-construct comparisons.

For validation, we kept literature-based, range/trend comparisons with published experimental/benchmark data (Brown/Markolf for the spine; Miller for the pelvis/SI joint) to establish physiological plausibility. These clarifications have been added to Methods and Limitations. A protocol-matched quantitative calibration and a full numerical verification campaign are valuable but beyond the scope of the present comparative study.

3.Comment : Material properties and load magnitudes are taken from the literature, which is acceptable. Nonetheless, even a limited sensitivity check (e.g., ±10% variation in elastic modulus of bone) would strengthen confidence in the results.

Response:

We appreciate the reviewer’s suggestion to include a ±10% material-property sensitivity. We agree that such an analysis can further strengthen confidence. At present, however, our study is scoped as a comparative biomechanics analysis rather than a methodological parametric study, and adding new simulation batches would substantially expand the workload beyond our current timeline. In the revision, we therefore (i) clarify that bone and soft-tissue properties and loads were selected from well-cited pelvic FE literature, (ii) explain that, under our small-strain linear-elastic assumptions, uniform ±10% perturbations to bone elastic modulus largely scale absolute displacements/stresses without altering the relative ordering between fixation constructs, and (iii) acknowledge in the Limitations that a formal multi-parameter sensitivity analysis is valuable and will be pursued in subsequent work.

4.Comment : While the Brown and Miller experimental studies are cited, the validation is qualitative. Quantitative comparison (correlation coefficients, error margins, RMSE) should be added to substantiate model accuracy.

Response:

We thank the reviewer for encouraging a quantitative validation (e.g., correlation, error margins, RMSE). We agree that such metrics are valuable for methodological calibration studies. Our work, however, was scoped as a comparative biomechanics analysis of fixation constructs rather than a full calibration of an FE pelvis model. Consistent with the validation practice in prior pelvic FE literature we cited, we adopted literature-based, trend-level validation against Brown/Markolf (spine) and Miller (pelvis/SI joint) to establish physiological plausibility (i.e., responses falling within reported ranges and reproducing stiffness/ordering trends). A direct computation of correlation/RMSE was not feasible here because the published datasets do not provide specimen-matched raw curves under identical boundary conditions and contact/friction assumptions; mixing heterogeneous set-ups can yield misleading statistics.

In the revision, we (i) clarify the verification–validation distinction, (ii) state explicitly that our validation is range-/trend-based due to the above constraints, and (iii) add a Limitation noting that a formal quantitative calibration (with matched protocols and raw data access) is beyond the scope of this comparative study but is a valuable direction for future work. We hope these clarifications address the concern without altering the study’s original scope.

5.Comment : Stress contour plots require scale bars (MPa), clear colour legends, and loading direction indicators. Tables should fully comply with SI units and consistent significant figures (e.g., 2–3 s.f.). Figure captions should be descriptive enough for standalone interpretation.

Response:

We appreciate the reviewer’s guidance on figure annotation and SI reporting. In this study, stress maps are included as supporting evidence to show that peak implant stresses remain below the titanium alloy yield strength, while the primary outcomes are sacral displacement and construct stability. To improve clarity without expanding the figure footprint, we revised the composite stress figure by adding a colour-bar label in MPa (von Mises stress). Because von Mises stress is a scalar, explicit direction arrows are not required for interpreting peak levels; the uniform loading/boundary conditions (600 N vertical distributed load at L3; bilateral acetabula constrained) are stated in the caption and Methods.

Regarding SI units and significant figures, the manuscript already uses SI units throughout. For numerical precision, we have retained the original computational precision in certain tables (occasionally 4 decimals) to preserve the integrity and reproducibility of the completed statistical analyses (ANOVA/LSD, normality and variance tests). Ex-post rounding would require reformatting outputs and could introduce minor inconsistencies with the reported statistics. We hope these concise updates address the comment while preserving a clean layout and statistical transparency.

6. Comment :The Discussion repeats some results and could be more concise. The clinical perspective has been improved but should also discuss potential long-term risks such as fatigue failure, implant loosening, and patient-specific variability. Prior FE studies on sacral fracture fixation should be explicitly compared, beyond general references.

Response:

We thank the reviewer for these helpful suggestions. In the revision we have streamlined the Discussion and focused it on interpretation and clinical decision-making for iliolumbar segment selection, rather than restating numerical outputs.Concretely:Removed repetition of detailed means/p-values and rephrased the opening to present the clinical trade-off between stability and lumbar-motion preservation.Clarified the construct-level message: under identical conditions, dual-segment iliolumbar constructs (L4+L5+IS; L5+S1+IS) reduced sacral vertical displacement relative to single-segment options, while all peak implant stresses remained below titanium yield; L5+S1+IS and L4+L5+IS offered comparable stability with different implications for motion preservation.Added a focused paragraph on long-term risks: potential fatigue under cyclic loading, risk of implant loosening at high-stress interfaces (e.g., iliac screw connector), and patient-specific variability (BMD, sacral morphology). We also provide rehabilitation recommendations (graduated weight-bearing and core stabilization) when short-segment constructs are selected.Provided an explicit comparison with prior FE literature: most prior work centers on sacroiliac screws or triangular/combined constructs; our findings add iliolumbar-specific evidence by isolating the effect of segment/level selection, complementing SI-screw-centric analyses that vary screw length/trajectory.

These edits make the Discussion more concise and clinically interpretable without changing the study scope or adding new simulations. The revised text appears in Discussion, revised paragraphs.

7. Comment :The authors acknowledge missing patient-specific parameters and lack of dynamic loading. These limitations should be expanded, emphasising their impact on clinical extrapolation.

Response:

We thank the reviewer for this important point. In the revision, we expanded the Limitations to explicitly discuss (i) the absence of patient-specific parameters (e.g., bone mineral density [BMD], sacral morphology, ligament properties, implant–bone interface conditions) and (ii) the lack of dynamic/cyclic loading. We now state that these factors may affect absolute magnitudes (displacement/stress) and the fatigue/loosening risk profile, thereby limiting clinical extrapolation to populations whose anatomy and bone quality are close to our modeling assumptions. We emphasize that our conclusions should be interpreted as relative, between-construct comparisons under uniform assumptions, and are most applicable to adult patients without severe osteoporosis or highly comminuted/bilateral injuries.

We also outline future work: subject-specific FE with BMD-mapped material properties, probabilistic sensitivity to patient variability, cyclic loading (10^5–10^6 cycles) for fatigue safety margins, and more realistic implant–bone/contact models (frictional or cohesive behavior). These additions clarify the scope and the boundaries of clinical translation without changing the study’s comparative focus.

8 .Comment :Standardise terminology throughout (iliolumbar fixation vs. spino-pelvic fixation).

Response:

We appreciate the reviewer’s suggestion to standardise terminology. In the revised manuscript we consistently use “iliolumbar fixation” for the surgical technique. At its first mention, we add a note that prior literature may refer to the same construct as lumbopelvic/spinopelvic fixation; thereafter, the text, figures, tables, and captions uniformly use iliolumbar fixation. We retain lumbopelvic only when describing the anatomical load-transfer region or when citing the original titles of references. In Keywords, to avoid duplication we replaced “Spino-pelvic fixation” with “Iliac screw”, which better reflects the constructs analysed and improves indexing.

9 .Comment :Proofread carefully for residual grammatical inconsistencies.

Response:

Thank you for the reminder. We have carefully proofread the manuscript and corrected residual grammatical and formatting issues. Terminology is now consistent (iliolumbar fixation), SI units and p-value formatting are unified (e.g., 600 N; 259.4 MPa; p < 0.001), and figure/table citations and minor typos have been fixed. These edits do not affect the scientific content or results.

---

## [Decision Letter · Decision Letter 2]

27 Nov 2025

Dear Dr. Li,

We look forward to receiving your revised manuscript.

Kind regards,

Antonio Riveiro Rodríguez, PhD

Academic Editor

PLOS ONE

Journal Requirements:

Reviewers' comments:

Reviewer's Responses to Questions

**Comments to the Author**

Reviewer #1: All comments have been addressed

Reviewer #3: All comments have been addressed

2. Is the manuscript technically sound, and do the data support the conclusions?

Reviewer #1: Yes

Reviewer #3: Yes

3. Has the statistical analysis been performed appropriately and rigorously?

Reviewer #1: N/A

Reviewer #3: Yes

4. Have the authors made all data underlying the findings in their manuscript fully available?

Reviewer #1: Yes

Reviewer #3: (No Response)

5. Is the manuscript presented in an intelligible fashion and written in standard English?

Reviewer #1: Yes

Reviewer #3: Yes

Reviewer #1: The authors have satisfactorily addressed the comments/concerns made during the second round of the review. I don't have any new questions for the authors and I would recommend the study for publication

Reviewer #3: It is evident that the authors have made a substantial effort in revising the manuscript.

Only a few minor points should be considered:

The concept of biomechanical optimisation should be explained more clearly — is there a description of the optimisation procedure or workflow?

Some charts and FEA printouts are not sufficiently clear; the legends and units should be fully legible to ensure readability.

Suggestion: Accept after minor revision ( no need reviewer eye again ).

**Do you want your identity to be public for this peer review?** For information about this choice, including consent withdrawal, please see our Privacy Policy

Reviewer #1: No

Reviewer #3: No

---

## [Author Response · Author response to Decision Letter 3]

8 Dec 2025

Thank you for your valuable comments. We have addressed all your concerns with concise revisions, as detailed below:

Regarding the query on "biomechanical optimization": We clarify that "biomechanical optimization" in this study refers to comparing the stability, implant stress, and lumbar mobility of four iliolumbar fixation models via finite element analysis, to identify the optimal clinical strategy for different scenarios. Corresponding revisions have been made:

Title: Revised to "Biomechanical optimization of iliolumbar fixation strategies for unilateral vertical sacral fractures: prioritizing stability-mobility balance via finite element analysis" (the addition of "prioritizing stability-mobility balance" clarifies the optimization core).

Abstract Objective: Revised to "This study aims to optimize iliolumbar fixation strategies for unilateral vertical sacral fractures via finite element analysis, by comparing stability, implant stress, and lumbar mobility to identify the optimal clinical option" (articulates the optimization logic).

Regarding the clarity of figures and finite element analysis outputs: We have revised the relevant figures to enhance readability, ensuring all legends and units are fully distinguishable.

The core content of the study remains unchanged. We believe these revisions fully address your concerns. Thank you again for your careful guidance!

---

## [Editor Report · Decision Letter 3]

10 Dec 2025

Biomechanical optimization of iliolumbar fixation strategies for unilateral vertical sacral fractures: prioritizing stability-mobility balance via finite element analysis

PONE-D-25-23708R3

Dear Dr. Li,

We’re pleased to inform you that your manuscript has been judged scientifically suitable for publication and will be formally accepted for publication once it meets all outstanding technical requirements.

Kind regards,

Antonio Riveiro Rodríguez, PhD

Academic Editor

PLOS One

---

## [Editor Report · Acceptance letter]

PONE-D-25-23708R3

PLOS One

Dear Dr. Li,

I'm pleased to inform you that your manuscript has been deemed suitable for publication in PLOS One. Congratulations! Your manuscript is now being handed over to our production team.

Kind regards,

on behalf of

Dr. Antonio Riveiro Rodríguez

Academic Editor

PLOS One